# *Taenia solium* excretory secretory proteins (ESPs) suppresses TLR4/AKT mediated ROS formation in human macrophages via hsa-miR-125

**Naina Arora[1¤], Anand K. Keshri[1], Rimanpreet Kaur[1], Suraj S. Rawat[1], Rajiv Kumar[2], Amit Mishra[3], Amit Prasad** [1] *

**1** School of Biosciences and Bioengineering, Indian Institute of Technology Mandi, Mandi, Himachal Pradesh, India, **2** Biotechnology Division, CSIR-Institute for Himalayan Bioresource Technology, Palampur, Himachal Pradesh, India, **3** Cellular and Molecular Neurobiology Unit, Indian Institute of Technology Jodhpur, Rajasthan, India

¤ Current Address: Current affiliation: Division of Biomedical Sciences, School of Medicine, University of California Riverside, Riverside, California, United States of America

* amitprasad@iitmandi.ac.in

**Data Availability Statement:** All relevant data used in this study are included in the manuscript and supplementary files. The miRNA data is publicly

## Abstract

### Background

Helminth infections are a global health menace affecting 24% of the world population. They continue to increase global disease burden as their unclear pathology imposes serious challenges to patient management. Neurocysticercosis is classified as neglected tropical disease and is caused by larvae of helminthic cestode *Taenia solium*. The larvae infect humans and localize in central nervous system and cause NCC; a leading etiological agent of acquired epilepsy in the developing world. The parasite has an intricate antigenic make-up and causes active immune suppression in the residing host. It communicates with the host via its secretome which is complex mixture of proteins also called excretory secretory products (ESPs). Understanding the ESPs interaction with host can identify therapeutic intervention hot spots. In our research, we studied the effect of *T. solium* ESPs on human macrophages and investigated the post-translation switch involved in its immunopathogenesis.

### Methodology

*T. solium* cysts were cultured *in vitro* to get ESPs and used for treating human macrophages. These macrophages were studied for cellular signaling and miR expression and quantification at transcript and protein level.

### Conclusion

We found that *T. solium* cyst ESPs treatment to human macrophages leads to activation of Th2 immune response. A complex cytokine expression by macrophages was also observed with both Th1 and Th2 cytokines in milieu. But, at the same time ESPs modulated the

available at https://www.ncbi.nlm.nih.gov/geo/query/acc.cgi?acc=GSE232005.

**Funding:** AP is supported through research grants BT/PR26841/MED/122/121/2017 Department of Biotechnology, Government of India, New Delhi and ECR/2016/000817/LS from Science Engineering Research Board (SERB), New Delhi. The funders had no role in study design, data collection and analysis, decision to publish, or preparation of the manuscript.

**Competing interests:** The authors have declared that no competing interests exist.

macrophage function by altering the host miR expression as seen with altered ROS activity, apoptosis and phagocytosis. This leads to activated yet compromised functional macrophages, which provides a niche to support parasite survival. Thus *T. solium* secretome induces Th2 phenomenon in macrophages which may promote parasite's survival and delay their recognition by host immune system.

## Author summary

This article discusses the research conducted on the effect of *Taenia solium* excretory secretory proteins (ESPs) on human macrophages and its role in the immunopathogenesis of Neurocysticercosis (NCC). NCC is a neglected tropical disease caused by larvae of *T. solium*, which infect humans and localizes in the central nervous system. The larvae cause NCC, a leading cause of acquired epilepsy in the developing world. The study found that *T. solium* ESPs induce a Th2 immune response in macrophages, which may promote parasite survival and delay their recognition by the host immune system. The research further showed that ESPs modulate macrophage function by altering host miR expression, leading to activated yet compromised functionality that provides a niche for parasite survival. This study provides insights into the interaction of ESPs with the host, revealing therapeutic intervention hot spots to manage NCC and other helminthic infections. In the absence of suitable animal model for NCC, our study elucidates important and novel finding of parasite mediated suppression of host immune system.

## 1. Introduction

More than a quarter of world's population is at risk of helminthic infections, which cause a significant disease burden, associated disabilities and stress to health care system. Chronic helminthic infections skew the immune response towards T-helper 2 (Th2) by altering the early immune responses generated by macrophages and dendritic cells [1].

Macrophages are front line warriors of the immune system and play a crucial role in defining the immune response as they influence several biological processes due to their plasticity [2]. They are a very versatile class of immune cells that can adapt to and differentiate into different subsets depending on the environmental stimuli. These heterogeneous immune cells have been described for their different activation states: "classically activated" M1 macrophages, "alternatively activated" M2 & M2 subtypes macrophages, regulatory macrophages and less defined subsets like tumor associated macrophages [3,4]. Since macrophages encounter and recognize a wide range of insults classified as "Danger Associated Molecular Patterns" (DAMPs) & "Pathogen Associated Molecular Patterns" (PAMPs) via exogenous or endogenous "Pathogen Recognition Receptors" (PRRs) for example Toll like receptors (TLRs), they can exhibit protective or pathogenic roles in different stages of an infection or a disease. The M2 macrophages are induced by Th2 cytokines like IL-4 and are required for tissue repair, wound healing, unregulated inflammatory and autoimmune response [5]. The role of helminths derived products in determining the macrophage polarization has been reported [6] but has not been yet fully defined, especially for cestode "*Taenia solium*". *T. solium* is a highly prevalent species of helminth infecting humans and swine [7]. The larvae infiltration to human central nervous system (CNS)/brain causes severe infection i.e., Neurocysticercosis (NCC), which leads to epilepsy. Understanding the ESPs interaction with the host may reveal

therapeutic intervention hot spots to manage NCC and other helminthic infections in a larger perspective.

NCC is an extremely widespread CNS associated infection and is the single most common cause of late onset of acquired epilepsy, more frequently in developing tropical countries [8–10]. As per World Health Organization (WHO) report (https://www.who.int/news-room/fact-sheets/detail/taeniasis-cysticercosis 2015), out of 50 million new epileptic incidences reported worldwide, approximately one third cases are due to NCC in endemic regions [11]. The WHO Food Borne Disease Burden Epidemiology Reference Group 2015 had identified "*T. solium* as a leading cause of deaths by food borne disease considerable to 2.8 million disability adjusted life years loss (DALYs)" (https://www.who.int/news-room/fact-sheets/detail/taeniasis-cysticercosis 2015)[11]. Although the endemicity of the disease is restricted to sub-tropical and tropical parts of low- and middle-income countries, incidences from developed countries are also often reported owing to frequent travel and immigration of individuals from endemic regions to non-endemic areas [12]. The clinical presentation of NCC patients is pleomorphic. The most common clinical presentation is recurrent epileptic seizures [13–15]. To date, the immune pathogenesis of this disease is unclear, due to complex biology of the parasite that includes the lack of knowledge about the role of different antigenic pools generated by this parasite, especially the parasite secreted antigens [15–17].

Unlike intra-cellular parasites which reside inside the host cells, *T. solium* is an extracellular parasite and parasite's secreted factors or secretome are predominantly involved in shaping host immunity [18–20]. The parasite secretory factors or excretory secretory products (ESPs) are the parasite's mode of communication with the host, and they interfere with host's immune-signaling mechanism and immune homeostasis. These ESPs constantly keep changing their composition as per the environmental stimuli received by the parasite to support its survival. The ESPs of *T. solium* are a complex mixture of proteins as they are constantly secreted by live metacestodes hence identifying them is a daunting task. Fortunately, the whole genome of *T. solium* had been published [21] and the WGA for secretome had identified 838 proteins as ESPs. Out of these 838 proteins, mRNA for 347 of them was identified and the KEGG analysis had identified 166 pathways that includes protein processing, pathways in cancer, focal adhesion, PI3K-Akt pathway, Wnt signaling, glycerophospholipid metabolism etc. [22]. These reports had helped us in our earlier studies where we had identified the different proteins present in *T. solium* secretome and kinome associated with different cellular processes like cell proliferation, cell adhesion, migration, and maturation [23–24]. This also established ESPs as a significant source of "immunogenic proteins" due to their availability to be processed and recognized by the host immune system. These findings had made the ESPs more lucrative for further investigation. Effects of various antigens like vesicular fluid, cyst wall or crude lysate of *T. solium* on immune cells have been reported [25]. However, the effect of ESPs antigens on macrophages has never been explored before, though such studies are essential for our understanding of helminth-host interaction. Hence, in this study we investigated the immune response of *T. solium* ESPs on human macrophages.

## 2. Material and methods

### 2.1 Ethics statement

All the work was performed after taking required approval from the Institute Ethics Committee and Institute Biosafety Committee of Indian Institute of Technology Mandi (Ref No. IITM/IEC(H)/2022/AP/P7). A formal written consent was obtained from each participant, and formal consent was obtained from the parent/guardian of child participants.

## 2.2 Isolation of excretory secretory proteins (ESPs)

The ESPs were made as previously described [23]. Briefly, *T.solium* cysts were isolated from naturally infected swine after sacrifice at local butcher house and washed thrice thoroughly in chilled PBS supplemented with 1% antibiotic-antimycotic (#1S40096, Invitrogen, USA). Cysts were cultured in RPMI 1640 supplemented with 1% antibiotic-antimycotic for 24hrs. After 24hrs, culture supernatant was collected and filtered through 0.45uM filter and this was considered as excretory secretory products (ESPs) and stored at -80˚C for further experiments.

## 2.3. Cell culture and treatment

The human origin cell lines U937 and THP-1 were purchased from the national cell repository at National Center for Cell Sciences (NCC), Pune, India. Cells were cultured as per guideline in Gibco RPMI 1640 (#11875093) cell culture medium supplemented with 10% Gibco Fetal calf serum (FCS). The cells were differentiated to macrophages using 10ng/mL phorbol 12-myristate 13-acetate (PMA) (#79346, Sigma-Aldrich, USA) for 24-hr, followed by a 24-hr rest period and treated with 20μg/ml of ESPs antigens for another 24-hr for all our experiments, until stated otherwise.

## 2.4. Extraction of RNA and QPCR

The treated cells were lysed with TRIzol LS reagent for total RNA extraction and stored at -80˚C till use. All the procedures were performed following protocols as described previously [26]. RNA was reverse transcribed with a commercial kit (#1708891, iScript, BioRad) to generate initial cDNA and the samples were stored at -20˚C until use. The list and sequence of primers used for QPCR of cytokines, TLRs and β-actin gene and annealing details used are given in S1 Table.

## 2.5. Cytokines protein quantification

**2.5.1. Cell based assay.** Cytokines quantification at protein level was also done using flow cytometer based Cytometric Bead Array with BD "CBA Human Th1/Th2/Th17 Cytokine Kit" (Cat No.560484) as per manufacturer's guideline. The samples were acquired on BD LSR Fortessa and analysis was performed on BD CellQuest software.

**2.5.2. Enzyme linked immunosorbent assay (ELISA).** ELISA samples were prepared as described previously (Singh et al., 2015) [26]. Cell culture media was collected, aliquoted and stored at -80˚C for further use. After estimating protein concentrations all the samples were diluted to adjust the final protein concentration of 500 μg/ml and equal amount of protein was subjected to ELISA using commercially available kits (Invitrogen, USA) as per manufacturer's instructions. The results were expressed as picograms of cytokine/mg (pg/mg), based on the standard curve sketched with standard provided with the kits.

## 2.6. Protein expression quantification

Expression of TLR, AKT, pAKT was performed at translation level by western blot. Primary monoclonal antibodies (β-actin #8457, TLR4 #14358 AKT #4691 pAKT#4056) were purchased from Cell Signaling Technology (USA) and used as per manufacturer's instruction. For pAKT/ AKT expression quantification, cells were treated with or without ESPs or LPS and stimulated with fMLP (N-formyl-Leu-Phe) or challenged with E.coli (#59880-97-6, Millipore Sigma). The blots were developed using ECL Western Blotting Substrate (Thermo scientific #32109) and imaged on an imager (Amersham Imager 600).

## 2.7. ROS measurement

**2.7.1 By Flow cytometer.** The ROS generation was quantified through flow cytometer too, using CellROX orange flow cytometry assay kit (C10443, Invitrogen, USA). In brief, $5x10^5$ cells were taken and incubated with and without ESP for the mentioned time. To prepare negative control, cells were treated with N-acetylcysteine (NAC) (1mM) for one hour to increase the antioxidant capability of the cell. For positive control tert-butyl hydroperoxide (TBHP) (200uM) was added to cells and incubated for 30–60 min. TBHP was also added to negative control post incubation with NAC. Once the controls were set up, the cells for treatment group and controls were stained with CellROX reagent (250uM) and incubated for 30–60 min at 37˚C and data was acquired through a flow cytometer (BD LSR Fortessa).

**2.7.2. By Cytochrome C assay.** To assess the superoxide production from macrophages, cytochrome-c assay was performed. In brief, $0.5x10^6$ cells were pre-treated with antigen, stimulated with 1uM fMLP for 5min and incubated with 1.5 mg/ml of Cytochrome C (from equine heart; #9007-43-6, Millipore Sigma) for 5 minutes at 37˚C. The reduction in Cytochrome C was recorded for each sample by measuring the absorbance at 550 nm every 30sec for 10 minutes.

## 2.8. Phagocytosis assay

Zymosan BioParticles (Invitrogen, Carlsbad, CA) were incubated with 10% human serum in PBS for 1 hour to opsonize. The differentiated macrophages cells were suspended in Gibco HBSS (#14175095) at a density of $1x10^7$/ml. The opsonized Zymosan particles and cells (treated with and without ESPs) were then incubated at a 1:50 ratio at either 37˚C or 4˚C with end-to-end rotation for 1 hour. "Extracellular fluorescence was quenched by adding trypan blue, and the phagocytosis index (PI) was calculated as the number of bioparticles engulfed by 100 macrophages" as described previously [27]. For each group >200 cells were counted.

## 2.9. In vitro bacterial killing assay

*Escherichia coli* (strain 19138; ATCC, Manassas, VA) and *Staphylococcus aureus* (strain 10390; ATCC, Manassas, VA) were freshly cultured overnight and resuspended in PBS at an OD600 of 0.20. They were then opsonized with 10% human serum for 90 minutes at 37˚C in a water bath. The cells were first pre-treated with ESPs and AKT activator SC79 (#305834-79-1, Millipore Sigma) and then with bacteria at a ratio of 1:10 for *E. coli* and 1:20 for *S. aureus* for different time intervals of 0, 30, 60, and 120 minutes at 37˚C with intermittent shaking. Control group cells were pre-treated with PBS only. To lyse the cells, autoclaved $H_2O$ was added, and diluted aliquots were spread on LB agar. The plates were then incubated overnight at 37˚C, and the colony-forming units (CFU) were counted. A bacterial suspension without any cells was used as an input control.

## 2.10. Extraction of miR and microarray

Total miR was extracted from cultured cells by miRNA easy mini kit (Cat No./ID: 217004, Qiagen, USA) as per the manufacturer's protocol and was stored at -80˚C till use. In brief, the cells were lysed using lysis buffer (500ul) and incubated at RT for 5 min, followed by addition of chloroform (140ul), incubated at RT for 2–3 mins and centrifuged at 12000g for 15 mins, 4˚C. Aqueous phase was taken to a new tube and mixed with 1.5vol of 100% ethanol. The resulting mix was transferred to the spin column and centrifuged at 12000g for 15s. Flow-through was discarded, spin column was washed with RWT once (700ul), followed by RPE (500μl) twice for 15s and 1min and RNA was eluted in 11ul of RNase free water. The RNA obtained was

quantified and checked for RIN on 2100 Bioanalyzer Instrument (#G2939BA, Agilent) (small RNA chip #5065–4413) value before proceeding to microarray (RIN >9).

For miR expression, a microarray experiment was performed on GeneChip miRNA 4.0 Array (Cat No. 902412). The input miR taken was 400ng and the poly(A) tailing, flash tag labeling and hybridization cocktail were done as described in manufacturer's protocol for FlashTag Biotin HSR RNA labeling kit (Affymetrix 901910). Sample to chip hybridization was done overnight, the hybridization cocktail was removed, and the array was stored in array holding buffer, followed by washing and staining the array using fluidics console after which the array was loaded into the array scanner.

For miR relative expression, the extracted miR was reverse transcribed using miScript II RT Kit, Qiagen (#218161) using HiSpecBuffer (present in the kit). The cDNA was used to study relative expression using Qiagen miScript SYBR Green PCR Kit (#218075) that comes with universal primer. Primer sequences for miR are given in S2 Table. RNU6 Qiagen primer assay, miScript Primer Assay (#218300) was used for endogenous control.

## 2.11. miR Transfection

mirVANA miR mimics were ordered from Ambion Thermo fisher (hsa-miR-1246 **ID**: MC13182, hsa-miR-6820-5p **ID**: MC26881, hsa-miR-3201 **ID**: MC16523, hsa-miR-125a-5p **ID**: MC12561). The 5 nmol lyophilized miR mimic was resuspended in nuclease free water to make 10uM working stock. We transfected $1x10^6$ cells with 30pmol of miR mimic using Neon NxT Electoporation system as described earlier [28] In brief the cell and miR mimic mixture were loaded in transfection tip and electroporated at 500 volts with five pulses for 10ms according to manufacturer's protocol. The cells and miR mixture were plated and incubated for minimum of 6 hours prior to treatment (for cell attachment) at 37˚C and 5% CO2, post incubation the cells were washed with PBS to remove dead cells and stimulated with ESPs/LPS as per the test group and further incubated for 24 hours. After treatment, cells were washed, scrapped, and pelleted down for western blotting or RNA extraction followed by quantitative gene expression as explained above.

## 2.12. Statistical analysis

The data here presented as "mean ± SD" from at least three independent experiments with three technical replicates each, if specified otherwise.

One way ANOVA F test was applied to analyze variations in cytokine concentrations among cell groups stimulated with ESPs, LPS and treatment control. The comparison for two groups with three different stimulations was made employing "multiple comparisons using Bonferroni "T- test". Correlations were calculated using "Pearson's test". A "P value" of less than or equal to 0.05 was considered statistically significant. The comparisons of averages between groups were analyzed by "one way ANOVA", and the "Dunn Multiple Comparison Test" was further used to determine significant differences between groups at the significance level of P < 0.05 (*P < 0.05 and **P < 0.01) [29].

# 3. Results

## 3.1. T. solium ESPs induce both Th1/Th2 cytokines

ESPs are immunogenic and immune modulatory as they can elicit immune response and skew the host immune response for its persistence and survival. Th2 response is more widely studied and associated with parasite clearance, but local chronic inflammation with parasite is associated with less clearly understood Th1 response. Since NCC infection can transition from

asymptomatic phase predominated by Th2 response to symptomatic phase with more pronounced Th1 response we studied both Th1 & Th2 cytokines in response to ESPs to identify the pointers associated with activation of respective immune response. The Th1/Th2 cytokines were measured at transcription (by qPCR) and translation level (by Cytometric bead assay). We found significantly elevated levels of Th1 cytokines like TNF-α, IL-1β & IL-6 along with signature Th2 cytokines IL-4 & IL-10 by qPCR (Fig 1A). The same heterogenous cytokines response was observed at protein level also, with IL17A, TNF-α, IL-1β, IFN-γ and IL-6 along with signature Th2 cytokines IL-4 and IL-10 significantly high in ESPs treated macrophages (Fig 1B). The progression of cytokines spectrum is given in **Fig E in** S2 File, showing peak for TNF-α, IL10 and IL-6 at 12 hr post stimulation.

## 3.2. ESPs induce polarization of macrophages to Galectin3 subset of macrophages

Helminth-derived products play vital role in deciding the fate of monocytes and it is reported that they promote alternatively activated macrophage (AAM) lineage [6,30]. The heterogenous cytokine led us to ask the macrophage polarization status, as there can be various subsets of M1/M2 macrophage depending on the stimulus. The cells treated with ESPs were evaluated for M1/M2 macrophage lineage marker and we found significantly high expression of Galectin 3(also known as Mac-2 previously) compared to control (Fig 2). Galectin 3 macrophages have been described for their role in phagocytosis and clearance of pathogens [31]. A study on murine NCC had reported infiltration of CNS with M2 macrophages expressing Galectin 3 and Galectin 3 deficiency correlated with parasite severity in CNS [32,33]. These findings along with our observation of Galectin 3 expression suggest parasite secreted products leads to M2 polarization of macrophages, which is crucial in regulating inflammation. We also observed higher expression of Arginase, though this was not significant (p = 0.063).

## 3.3. *T. solium* ESPs effect on TLRs

The TLRs play a key role as the upstream signaling molecules of inflammation and several of them have been associated with NCC pathogenesis. In a murine model study of NCC, all the TLRs (from 1–9) were found to be abundantly expressed in microglial cells except the TLR 5 [34]. But the expression of the TLRs induced by ESPs antigens in macrophages has not been reported yet. After stimulation of macrophages with ESPs, we also observed significantly high expression of genes for all the TLRs (1–9) except for the TLR 7, Fig 3. This observation was interesting since both TLR7 and TLR 8 binds with ssRNA.

The diverse make-up of ESPs was rich in ligands for all the TLRs. We specially checked the expression of TLR4 by doing western blot too as it has been reported that TLR4 is related with symptomatic NCC [35] and has been associated with immune pathogenesis of NCC. Interestingly, we noticed no change in TLR4 expression at protein level (Fig 4) though we had observed higher expression of TLR4 genes by qPCR (Fig 3).

## 3.4. ROS production, reduced AKT activity and bacterial killing capability

TLR4 has been found to be directly involved in the activation of master regulatory molecule Akt by converting to pAkt, and pAkt is responsible for generation of reactive oxygen species (ROS) [36–37]. The ROS are indispensable mechanism of the innate immune response against infections and immensely contribute to bacterial killing by an immune cell [27]. To establish the effect of ESPs on TLR4/pAkt mediated ROS production, we first measured ROS production then measured pAkt/Akt by western blot in. The CellROX ROS assay showed ESPs antioxidant potential is very similar to well-known antioxidant NAC (Fig 5A). The cytochrome-c

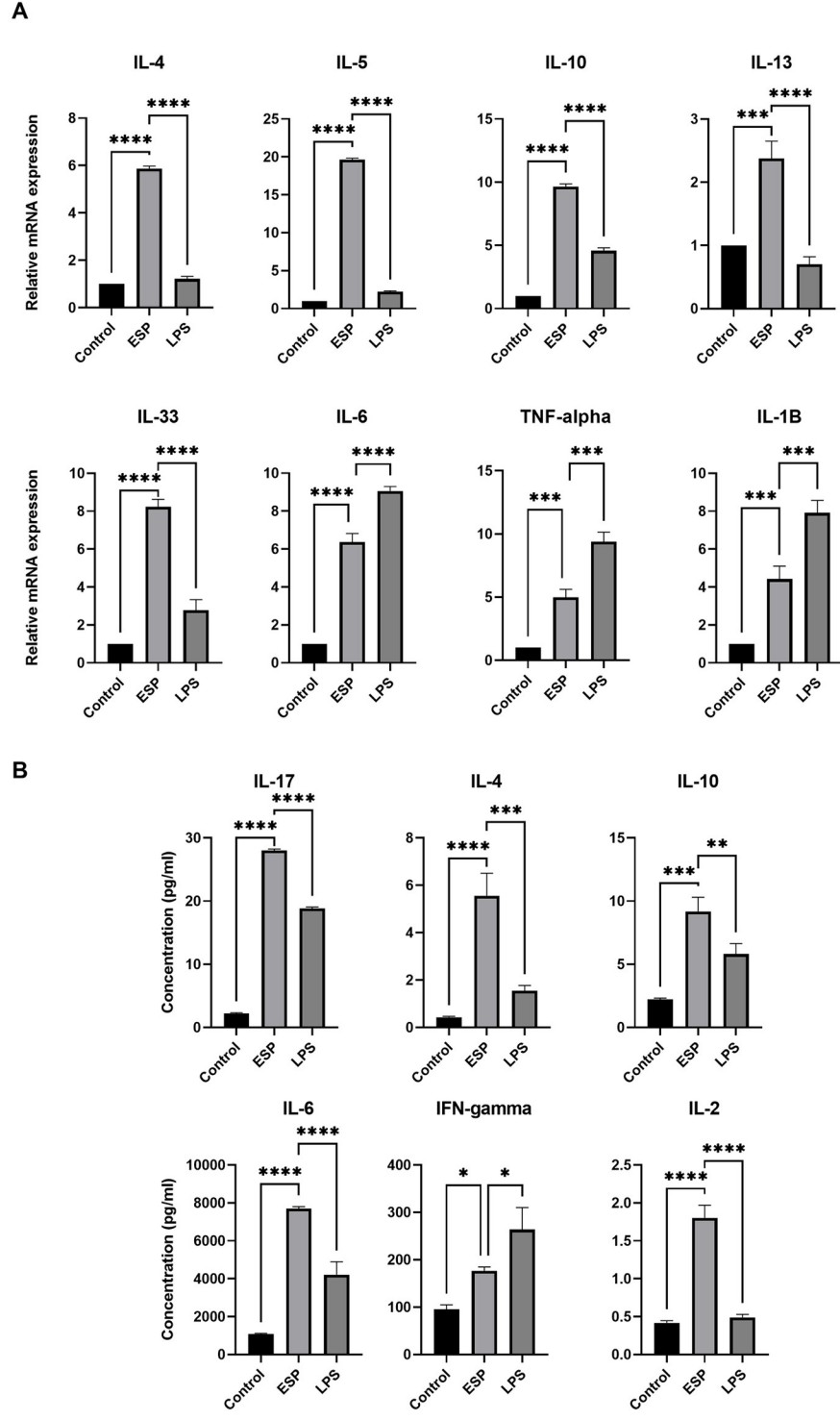

**Fig 1. *Taenia solium* derived ESPs activate both Th1 and Th2 cytokines in macrophages.** (A) Relative gene expression of cytokines from macrophages treated with ESPs, normalized to β-Actin as housekeeping gene. (B) Absolute quantification of secreted cytokines in the milieu of *T. solium* ESPs treated macrophages measured using cytometric bead assay.

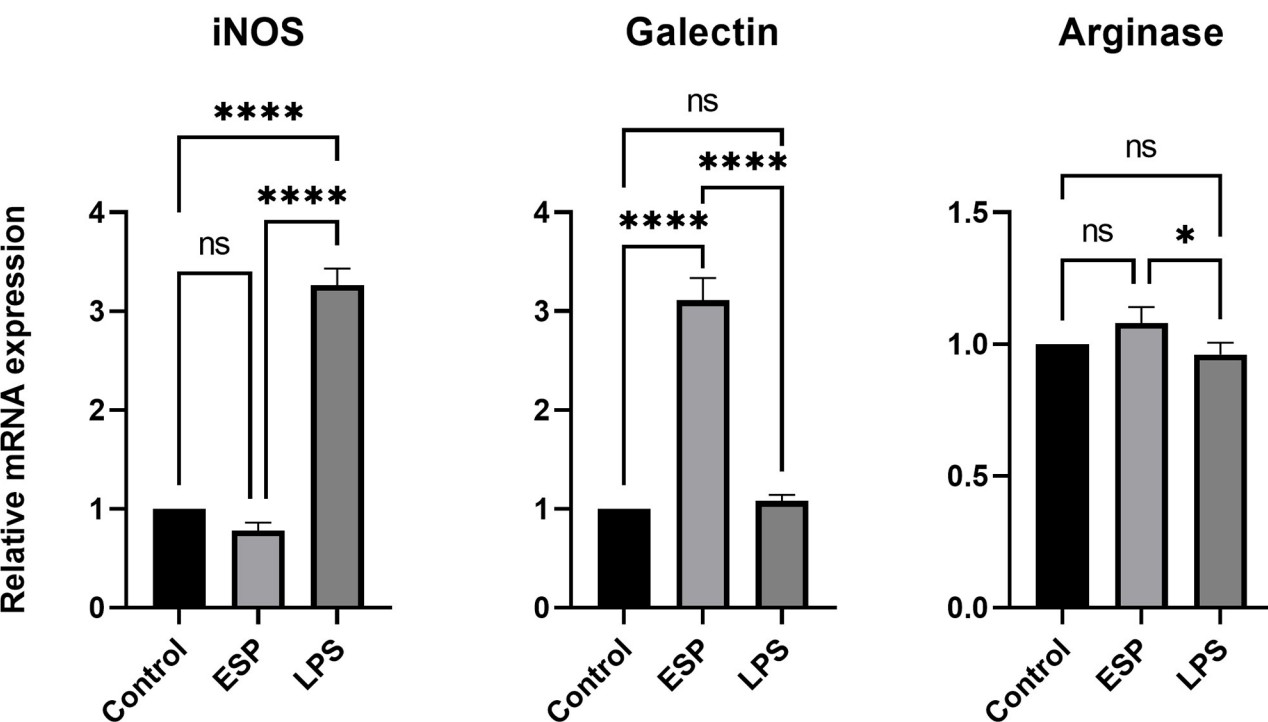

**Fig 2. *T. solium* ESPs polarize macrophages to Galectin 3-subset of macrophages phenotype.** Relative expression of macrophages markers normalized to housekeeping gene β-Actin.

reduction assay was also performed on ESP treated cells. This assay measures the total amount of ROS produced by a cell, and we noticed significantly less ROS production by ESPs treated cells (Fig 5B).

The treated cells tested for ROS were also quantified for pAKT/AKT activity and we found reduced pAKT in macrophages incubated with ESPs compared to LPS or negligible increase compared to control, when stimulated with a strong chemoattractant fMLP (Fig 6A). We checked the Akt activation in biological setting when they were kept for phagocytosis of *E. coli* and noticed reduced pAkt (Fig 6B). This suggested that ESPs treated macrophages might have defects in their bacterial killing capability and cell survival which is contrary to Galectin 3 expressing macrophages since they are known for their phagocytic capability. To confirm our observation that this reduced ROS is due to less Akt activity, we evaluated these treated macrophages for their bacterial killing capability by following previously published protocol [27]. We observed significantly (p<0.001) more CFUs in ESPs treated cells (163) when compared to control (102) (Fig 6C). We used small molecule SC79 (Merck Millipore, CAS 305834-79-1; [37], an Akt activator along with ESPs and we saw recovery of phenotype. The macrophages treated with ESPs and SC79 had tentatively lower number of CFUs (Fig 6C). This restoration of bacterial killing capability by Akt activator SC79 in ESPs treated macrophages confirms the role of ESPs in decreasing Akt activity.

This reduced ROS and diminished bacterial killing capability of macrophages on ESP treatment could stem from decreased phagocytic capability too as pAkt is also responsible for making f-Actin [38]. To assess this, we counted the number of zymosan-bioparticles engulfed by individual macrophages via an in-vitro phagocytosis assay (Fig 6D). On average, we found 53 "serum-opsonized fluorescein-conjugated zymosan particles" were engulfed by 100 control

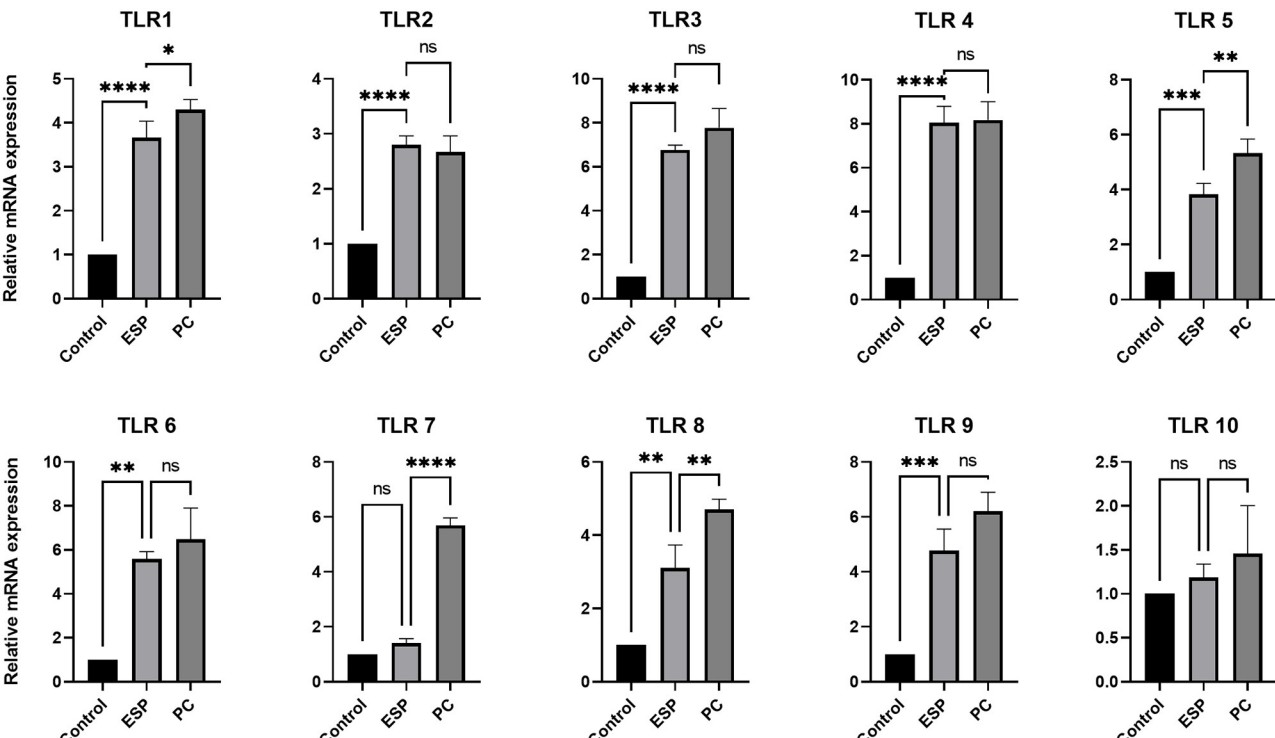

**Fig 3. Relative expression of TLRs in ESPs treated macrophages estimated.** PC- positive control, TLR agonists used for each of the TLRs from Human TLR agonist kit -InVivoGen Cat Code # tlrl-kit1hw (TLR1/TLR2 Agonist: Pam3CSK4, TLR2 Agonist: HKLM, TLR2/TLR6 Agonist: FSL-1, TLR3 Agonist: Poly(I:C)-HMW, TLR3 Agonist: Poly(I:C)-LMW, TLR4 Agonist: LPS-EK standard (LPS E.coli K12), TLR5 Agonist: FLA-ST standard (Flagellin S. typhimurium), TLR7 Agonist: Imiquimod, TLR8 Agonist: ssRNA40/LyoVec, TLR9 Agonist: ODN2006).

macrophages (phagocytic index) (Fig 6D). While in ESPs treated macrophages the number was significantly low i.e., 24 (p<0.001), suggesting a significant loss in phagocytic capability too. Akt activation has been associated with cell survival also as they give survival signal [27], hence we evaluated the ESPs treated macrophage for apoptosis however we could not find any difference (**Fig F in** S2 File).

### 3.5. Epigenetic control of TLR4 expression

Small non-coding RNAs (miRNA, siRNA and piRNA) are crucial regulators of gene expression and often execute their function by silencing target genes in a broad range of living forms including metazoans, fungi and plants [39–40]. The "miRs" or "miRNAs" belong to a class of small non-coding RNAs. They regulate gene expression of multiple gene targets within the same or distinct signaling pathways by directly binding to target based on sequence complementarity at 3′ untranslated region (3′ UTR) of target mRNA [41]. The studies related to role of miRNAs in helminthic infection are few, but has been explored previously [42]. To identify the miRs related with TLR4/Akt/inflammation axis we did micro-RNA microarray of ESPs treated THP-1 macrophages. The pixel intensity obtained was normalized and converted to .chp file which was analyzed using Affymetrix transcriptome analysis console. The microarray raw data is available at https://www.ncbi.nlm.nih.gov/geo/query/acc.cgi?acc=GSE232005 (GSE232005). The heat map of miR microarray is given in Fig 7A.

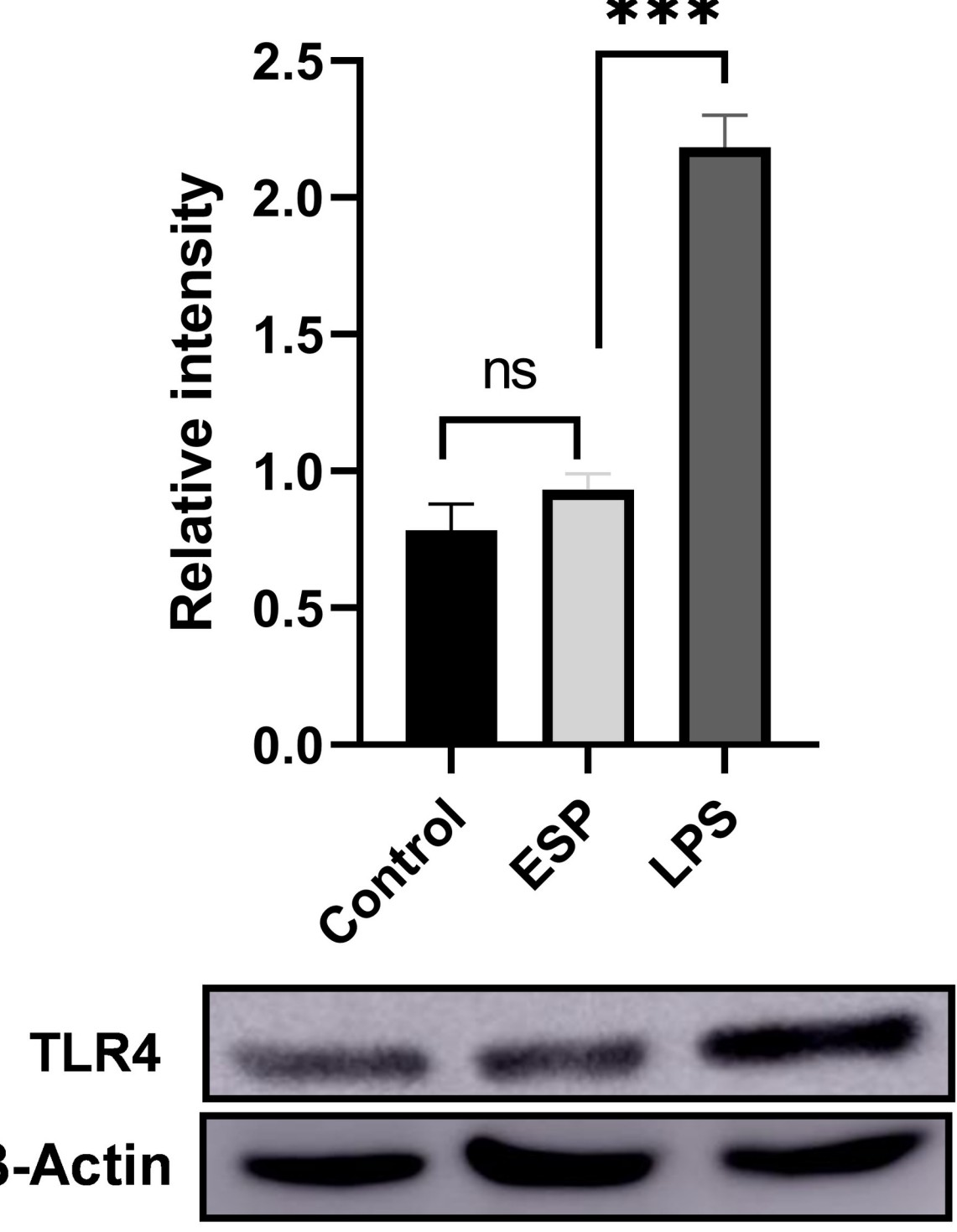

**Fig 4. Western blot analysis of TLR4 expression in macrophages treated with *Taenia solium* ESPs antigens.** The densitometry analysis of western blot bands was normalized to housekeeping protein β-Actin. The results shown are representative of three independent biological replicates.

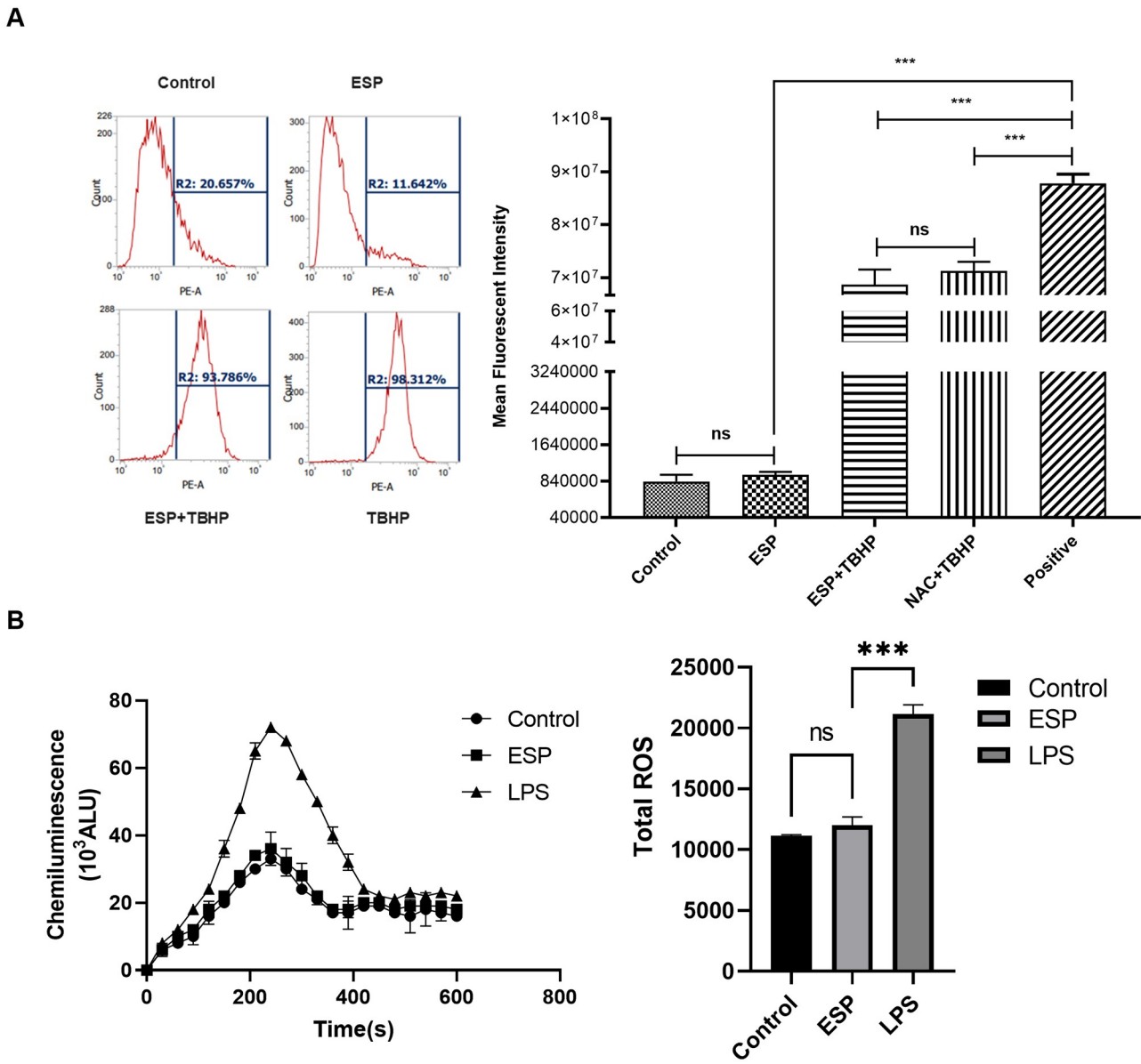

**Fig 5. Evaluation of ROS released by macrophages upon ESPs stimulation.** A) Human macrophages cells treated with ESPs+TBHP showed significantly less ROS when compared to cells treated with TBHP alone (positive control) in a flow cytometer based ROS assay (left) representative histogram plots of Cell ROX orange dye on flow cytometer, (right) qunatification of Cell ROX orange dye across the test groups B) The measurment of total ROS generation by Cytochrome C assay also confirmed reduced ROS in ESPs treated macropahges. Absorbance at 550 nm was measured 5 min after addition of cytochrome C for 10 mins at 30 sec interval (left). Total ROS respresented as bar graph from area under the curve (right).

From this list we selected miRs, whose expression had changed at least 1.5 folds and were searched in "Target scan" and "miRNet" for targets involved in TLR4/Akt/inflammation and further validated by qPCR. The miR interaction pathway was generated at MIENTURNET [43] (S2 Table). The qPCR of selected miRs validated our microarray data and we observed significantly high expression (p<0.005) of hsa-miR-19 and down regulation of hsa-miR-125a, hsa-miR-146a and hsa-miR155 in ESPs treated macrophages cells. It has been reported that hsa-miR-19 regulates Akt activity indirectly by regulating Phosphatase and tensin homolog

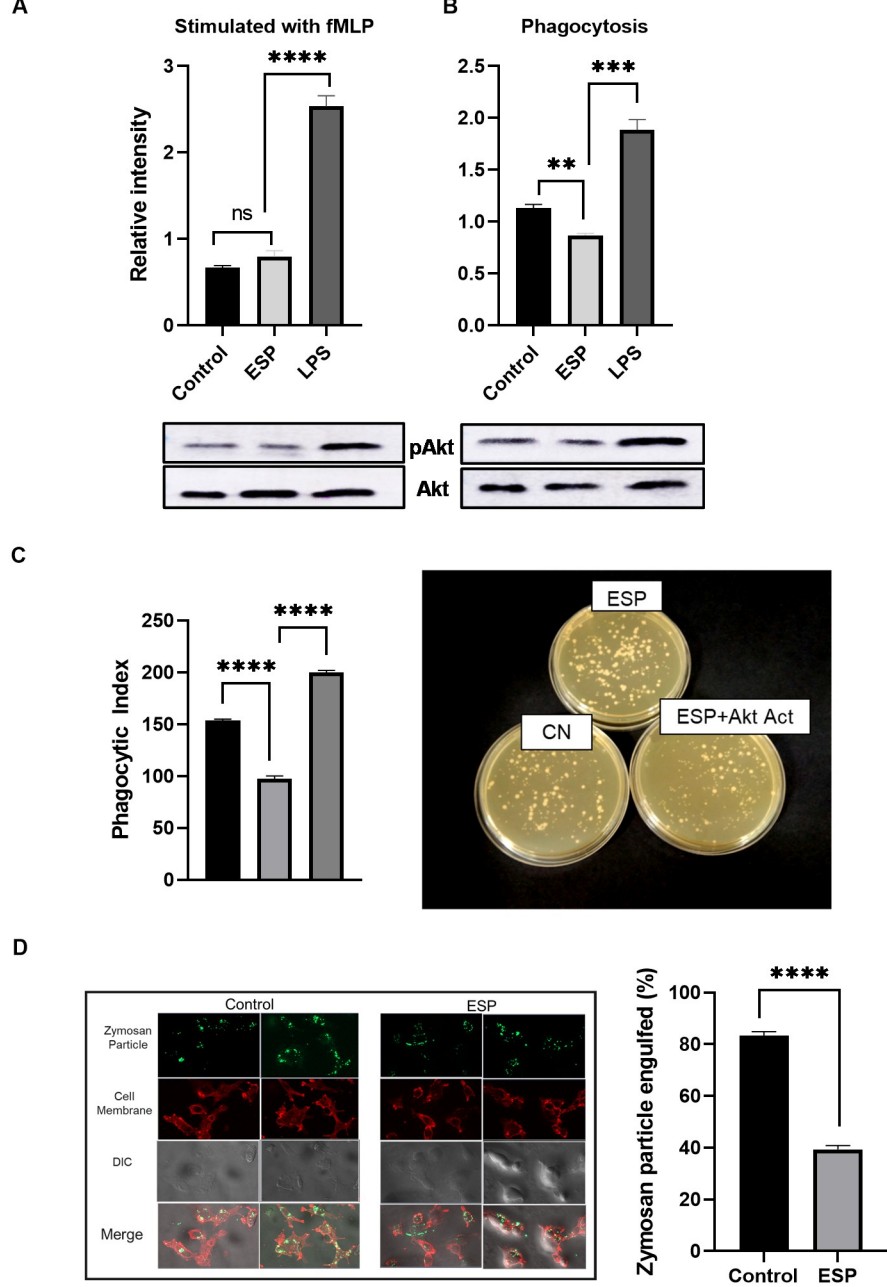

**Fig 6.** *Taenia solium* **derived ESPs treatment suppresses Akt activation in macrophages when stimulated with** A) Western blot quantification for pAKT/AKT in treated cells stimulated with fMLP B) Western blot quantification for pAKT/AKT in treated cells kept with *E.coli* for phagocytosis. C) *Taenia Solium* ESPs treated macrophages had reduced bacterial (*E. coli*) killing capacity, AKT activator SC79 rescued the lost phenotype in ESP treated cells. D) Macrophages treated with ESPs had less Zymosan bioparticles uptake, suggesting a decrease in phagocytotic capability as seen in flouresence microscopy (left) and phagocytic index (right). The results shown are representative of three independent biological experiments, in five randomly selected high power view per slide.

(PTEN) enzyme, it positively regulates the activity of PTEN and PTEN negatively regulates Akt activity [44]. We also noticed less pAkt in ESPs treated macrophages. In contrast to upregulated miR-19, we see downregulated miR-125 in microarray which is involved in Akt mediated apoptosis pathway. The expression profile of miRs in microarray strongly resonates with

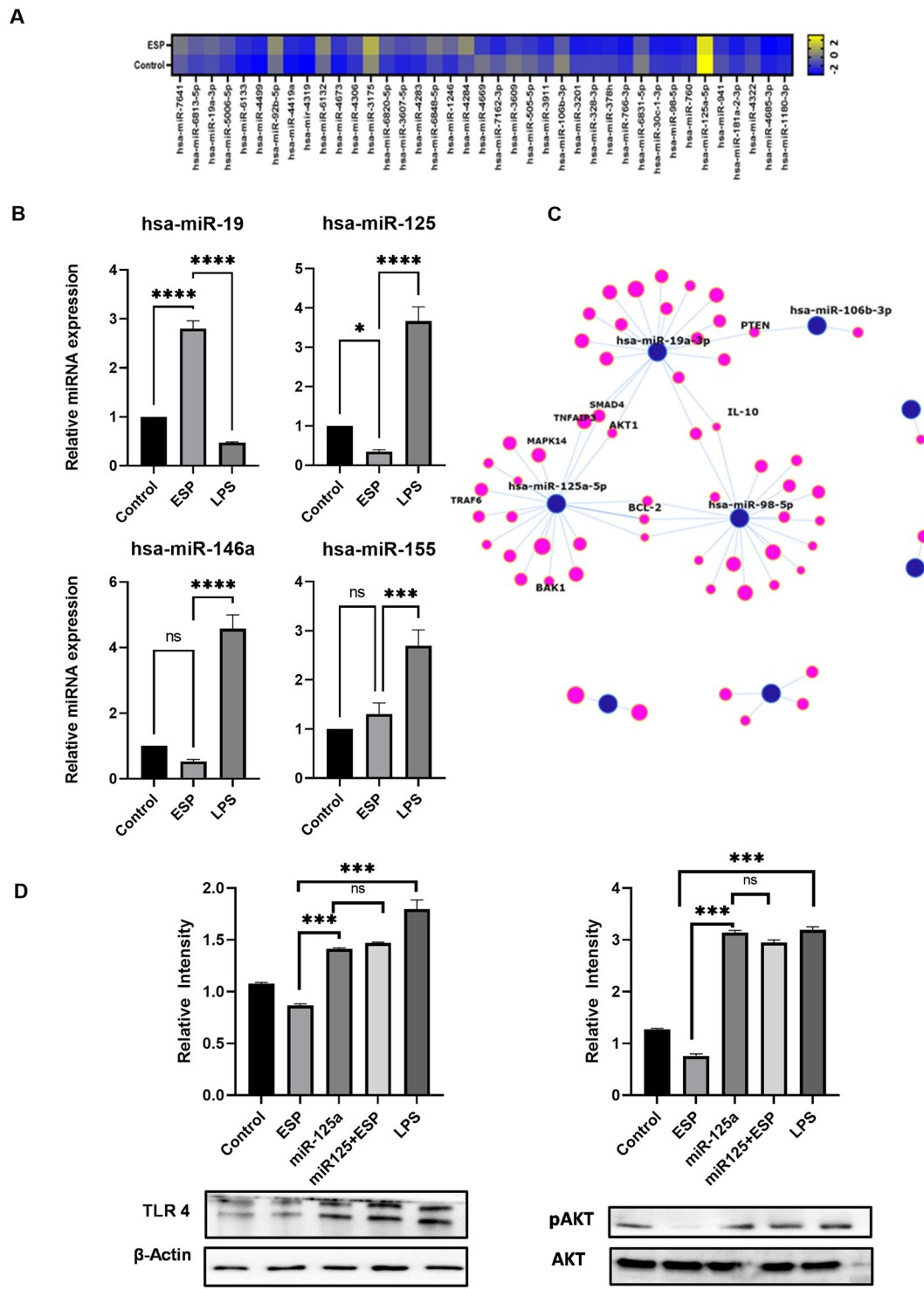

**Fig 7. *T. solium* ESP alters the epigenetic regulation in macrophages.** A) miR microarray of ESPs treated macrophages showed 57 differentially expressed miR, B) Validation of microarray identified miRs by qPCR. C). The representative graph showing miR network illustrating the interactions between miRs and their target genes (http://userver.bio.uniroma1.it/apps/mienturnet/) D) Western blot quantification of pAKT/AKT of macrophages transfected with hsa-miR-125 showed restoration of TLR4 expression in ESP treated macrophage.

the finding of our study and establishes that *T. solium* ESPs modulates at the post transcriptional level and impairs the macrophage's function. In myeloid cells, hsa-miR-146 controls the expression of NF-κB that in turn regulates NF-κB activation through a negative feedback loop thus restricting pro-inflammatory cytokines expression [45]. But we noticed down regulation of hsa-miR-146 and hsa-miR-155 which correlated to diminished inflammatory signaling, supported by downregulated Akt pathway. To further validate the role of miR mediated immune suppression by *T. solium* ESPs, hsa-miR-125 was transfected in ESPs primed cells to see if it restores the phenotype.

ESPs primed macrophages transfected with hsa-miR-125 restored the TLR4 expression as compared to positive control and had significant pAKT/AKT activity compared to ESP treated cells. ESPs impaired the pAKT/AKT activity in turn impairing the macrophage function and inflammation which is essential for elimination of larvae in initial stage of infection. The selection of miR-125 by the parasite may be attributed to the advantageous shared homology with mammalian miR-125, providing the parasite with a strategic edge in hijacking cellular machinery [46]. Our study suggests an immune modulatory effect of *T. solium* ESPs on these miRs and that sets up an environment to support parasite survival by disarming the inflammatory immune response. In another perspective, the dampened inflammation might be a tissue homeostasis mechanism required by the host to survive the parasite ESPs mediated insult.

## 4. Discussion

On exposure to pathogens or other stimulant, macrophages respond by triggering inflammatory responses. This response is initiated by signaling cascades that are activated downstream of TLR and cytokine receptors. Once activated, the macrophages initiate the production of pro-inflammatory cytokines and chemokines and other related proteins, the final outcome is controlled by transcriptional activity and associated epigenetic changes. The macrophages then migrate to the site of infection and eliminate the insult. Later, macrophages play a role in resolving the inflammation and preventing persistent inflammatory reactions that could damage surrounding tissues. Helminth derived antigens or the whole parasite can counteract pro-inflammatory responses generated during its infection and these antigens usually make immune response biased to Th2 type but also give some Th1 cytokines thus giving mixed cytokines profile [5–6,25]. This primarily Th2 type response assist host in parasite expulsion, tissue repair & regeneration, regulation of any inflammatory and autoimmune response. Some studies had been done involving *T. crassiceps* derived ESPs to identify the specific proteins [47–48] but this species infects rodents not human/swine, thus extrapolating these studies for clinically important *T. solium* infection is not an option. This is the first study so far, that had been taken to understand the role of *T. solium* derived ESPs in host macrophage modulation. Spolski et al (2002) looked for the immuno-modulating capability of *T. crassiceps* ES and noticed ES from early larvae but not late harvested larvae could suppress T cell proliferation *in vitro* and a decline in the production of IFN-γ & IL-4 [48]. Terrazas et al. 2013 reported defect in DC maturation, impaired Th1 cytokines production, and sensitivity towards TLR2 response that leads to Th2 polarization [49]. They had also reported a greater number of alternatively activated macrophages in mice models. Similarly, Chauhan et al (2014) also noticed reduced macrophages activation and Th2 polarization of myeloid cells when treated with soluble parasite ligands [50]. We also noticed both Th1 and Th2 cytokines elevated expression (IL-1β, TNF-α, IL-6, IL-4 & IL-10) along with higher expression of M2 macrophages markers. This shift in macrophage polarization towards M2 phenotype is essential to protect tissue damage from prolonged inflammation as M1 macrophages are directed to kill while M2 macrophages are directed to heal [51].

TLRs are encoded in the germline and act as PRRs, they belong to innate immune system. Their primary role is to recognize a wide range of microbial products, and they are expressed on both immune and non-immune cells. TLRs are crucial for the body's defense against pathogens [51,52]. In an earlier study on murine cysticercosis model, all the TLRs except the TLR 5 were identified [34]. We also noticed same pattern but of TLR7, which was non detectable. This may be due to difference in immune response in case of murine and human NCC. In the same model they found high expression of TLR 11–13 also, with TLR13 being the most expressive in all the cells and brain areas [53]. The TLR4 is one of the most studied receptors as it activates its downstream adapter protein MyD88 and nuclear factor κB (NF-κB) that leads to production of pro-inflammatory cytokines. It is especially important in NCC pathogenesis as TLR4 Asp299Gly and Thr399Ile alleles were found to be significantly correlated with the incidence and progression to symptomatic NCC [35]. This TLR4 signaling is important for macrophages to mount its effector function as its activation leads to production of inflammatory cytokines and ROS [36] while diminished TLR4 leads to M2 polarization of macrophages. The THP-1 derived macrophages treated with ESPs had shown increased expression of all TLRs (1–9) at transcription level, but we saw no difference in TLR4 expression at translation level. This suggested the post-transcription regulation of TLR4, and this diminished TLR4 is also making macrophage polarized to M2 and supported our observation of higher expression of M2 macrophage markers. The TLR4 also plays a crucial role in Akt activation by conversion of Akt to active pAkt by activated PI3K [54]. So, we looked for pAkt and as expected we found decreased pAkt in the presence of chemo-attractant or *E. coli* in ESPs treated macrophages compared to control or LPS treated cells (Fig 6). Earlier studies had shown Akt activation to be directly associated with ROS production [27,55]. Hence, we also investigated whether the reduced Akt can lead to alteration in superoxide production in ESPs treated macrophages and we found significantly less ROS in ESPs treated cells (Fig 5). A variety of Reactive Oxygen Species (ROS), like $H_2O_2$, HO, $O_2$ etc., are synthesized and released to the extracellular space and phagocytic vacuoles, where they are known to facilitate destruction of the pathogens [56]. This motivated us to look for the bacterial killing capability of ESPs treated macrophages, and consistent with reduced ROS we found a defect in their bacterial killing capability and when we used chemical Akt activator it restored the phenomenon, thus confirming that reduced killing was due to reduced Akt mediated ROS production (Fig 6).

Of late, there is an increased attention to understand the mechanisms and molecules of the helminth parasite's strong ability of immune modulation. But not much advances have been made in elucidating role of host immune cells miRs during helminthic infections, especially their role in the development, maturation & specialized function of immune cells [39]. The only literature available on helminths and miRs are about *Schistosoma spp*, *Fasicola spp* and *Clonorchis sinensis*. The effect of excretory-secretory antigens of *T. crassiceps* on inducing miRs was studied earlier and was found that these antigens repressed LPS-let-7i induction and diminishes inflammatory response in human dendritic cells [57]. Later, role of miR in TcES LPS-induced BMDM was explored and up-regulation of miR-125a-5p, miR-762 & miR-484 was reported. Implying TcES can change post-transcriptional regulation, thereby modulating proinflammatory responses in macrophages [6]. We also did microarray for miRs in macrophages after ESPs treatment and noticed difference in the expression of miRs, and qPCR for these miRs further confirmed upregulation of hsa-miR-19 and down regulation of hsa-miR-125a, hsa-miR-146a & hsa-miR155. Out of these miRs, miR-155, miR-223 & miR-146 have been described for their role in suppressing cytokine signaling and TLRs activation, and cytokine signaling via a negative feedback regulation loop by down-regulation of IL-1 receptor-associated kinase 1 (IRAK1) and TNF receptor-associated factor 6 (TRAF6) [58]. Thus,

corroborating and explaining our other observation of M2 polarization of macrophages and Th2 immune response with ESPs antigen.

In this study, we established that ESPs elicit mixed Th1/Th2 cytokines expression and make macrophages polarized towards Galectin 3-M2 phenotype. They also affect host miRs expression through which they suppress TLR4 expression, making these treated macrophages suppressed by diminished Akt activation and ROS production thus affecting their bacterial fighting capability. These findings about the role of ESPs on macrophages improved our understanding of host-parasite interaction and how the Taenia-released molecules suppress inflammation. These findings may open avenues for further research aimed at better diagnostic and treatment of NCC. In future, these molecules might find application in other inflammatory and autoimmune disorders. However, to reach that goal it will be imperative to explore the individual antigens identification and their immune modulation mechanisms adapted by them. Our results also suggest that the chronic NCC patients may be prone to other infections and this aspect should be explored by undertaking appropriate clinical study.

## Supporting information

**S1 File. Metadata for miRNA expression data from Taenia solium ESP treated human macrophages**
(XLS)

**S2 File. Fig A**. Flow chart outlining the method. **Fig B**. Cysts infested pork from naturally infected swine (a) and isolated cyst (b). **Fig C**. Species identification of Taenia by PCR, all our samples were positive only for *T. solium* and gave 984bp product specific for this species. **Fig D**. SDS-PAGE and silver staining of T. solium ESPs. **Fig E**. Quantification of progression of secreted cytokines by ELISA. **Fig F**. Apoptosis assay. The ESP treated cells showed no sign of apoptosis as compared to positive control. **Fig G**. Caspase activity. The ESP treated cells showed no sign of caspase 3 as compared to positive control. **Fig H**. Titration of ESP to establish diminished TLR4 expression and simultaneously decreased phosphorylation of Akt.
(DOCX)

**S1 Table. List of primers used in the study.**
(DOCX)

**S2 Table. List of primers used for miRNA validation in the study.**
(DOCX)

## Acknowledgments

NA and RK were supported by PhD student fellowship from the Indian Institute of Technology Mandi. SR acknowledges Department of Biotechnology, Gov. of India for salary support. AKK received salary from Council of Scientific and Industrial Research (CSIR), New Delhi.

## Author Contributions

**Conceptualization:** Amit Prasad.

**Data curation:** Naina Arora, Anand K. Keshri, Rimanpreet Kaur, Suraj S. Rawat.

**Formal analysis:** Naina Arora, Anand K. Keshri, Rimanpreet Kaur, Suraj S. Rawat.

**Funding acquisition:** Amit Prasad.

**Methodology:** Naina Arora.

**Project administration:** Amit Prasad.

**Validation:** Rajiv Kumar, Amit Mishra, Amit Prasad.

**Writing – original draft:** Naina Arora, Anand K. Keshri, Rimanpreet Kaur, Suraj S. Rawat, Rajiv Kumar, Amit Mishra.

**Writing – review & editing:** Naina Arora, Amit Prasad.

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
