## [Decision Letter · Decision Letter 0]

8 Aug 2023

Dear Dr. Prasad,

Thank you very much for submitting your manuscript "Taenia solium  excretory secretory proteins (ESPs) suppresses TLR4/AKT mediated ROS formation in human macrophages via hsa-miR-125" for consideration at PLOS Neglected Tropical Diseases. As with all papers reviewed by the journal, your manuscript was reviewed by members of the editorial board and by several independent reviewers. In light of the reviews (below this email), we would like to invite the resubmission of a significantly-revised version that takes into account the reviewers' comments. 

We cannot make any decision about publication until we have seen the revised manuscript and your response to the reviewers' comments. Your revised manuscript is also likely to be sent to reviewers for further evaluation.

Sincerely,

Andrew Scott MacDonald

Academic Editor

Cinzia Cantacessi

Section Editor

Reviewer's Responses to Questions

**Key Review Criteria Required for Acceptance?**

**Methods**

-Are the objectives of the study clearly articulated with a clear testable hypothesis stated?

-Is the study design appropriate to address the stated objectives?

-Is the population clearly described and appropriate for the hypothesis being tested?

-Is the sample size sufficient to ensure adequate power to address the hypothesis being tested?

-Were correct statistical analysis used to support conclusions?

-Are there concerns about ethical or regulatory requirements being met?

Reviewer #1: The study design is not appropriate for testing the hypothesis. The population size is not clearly described, nor does it seem sufficient.

Reviewer #2: -Are the objectives of the study clearly articulated with a clear testable hypothesis stated?

Yes

-Is the study design appropriate to address the stated objectives?

Mostly

**Results**

-Does the analysis presented match the analysis plan?

-Are the results clearly and completely presented?

-Are the figures (Tables, Images) of sufficient quality for clarity?

Reviewer #1: The figures are not of sufficient clarity for interpretation, and the legend do not offer enough details on the experimental design. The flow of the figures do not match the flow of the text, which can be confusing for the reader.

Reviewer #2: -Does the analysis presented match the analysis plan?

Yes

-Are the results clearly and completely presented?

Mostly

-Are the figures (Tables, Images) of sufficient quality for clarity?

Mostly

**Conclusions**

-Are the conclusions supported by the data presented?

-Are the limitations of analysis clearly described?

-Do the authors discuss how these data can be helpful to advance our understanding of the topic under study?

-Is public health relevance addressed?

Reviewer #1: The conclusions are not supported by the data presented.

Reviewer #2: -Are the conclusions supported by the data presented?

Yes

-Do the authors discuss how these data can be helpful to advance our understanding of the topic under study?

Yes

**Editorial and Data Presentation Modifications?**

Reviewer #1: (No Response)

Reviewer #2: (No Response)

**Summary and General Comments**

Reviewer #1: This manuscript address the role of ESPs in macrophage modulation. It is currently unclear if ESPs are indeed affecting polarization with the present datasets. Of note, it might be due to a poor presentation of the experimental design. The authors argue for a TLR4 reduction of ROS production through a miR dependent mechanism. However, if my understanding of the experimental design is correct ESP on its own is not a TLR4 agonist (as many other helminth products) and as such do not cause Akt phosphorylation. I failed to see how this is a decrease of anything ? If ESP combined with LPS, caused a decrease in Akt phosphorylation then the mechanism would make sense, as ESP seems to decreasing phagocytosis.

Please find detail comment sin the attached pdf.

Reviewer #2: The manuscript of Arora N et al. describes the function of Taenia solium excretory secretory proteins (ESPs) to supress TLR4/AKT ROS formation in human macrophages via has-miR-125. This study adds invaluable value to the field of how helminth can suppress host immune response to establish the infections. 

Major comments:

1. There are two bands of TLR4 in Figure 7D while Figure 4 has only one band, can authors please explain and show the full blot with labelling?

2. In 3.4. 

1) Akt inhibitor should be included as a control group. 

2) If the Taenia solium does affect microglia, a microglia cell line such as HMC3 could be used 

3. In 3.5., 

a miR-125 inhibitor could be added as miR-125 is suspected to be critical for the TLR4 and AKT expression

Minor comments:

1. methods do not provide details. please check all methods section

e.g. 

Line 139: name of antibiotics and concentration are missing?

Line 147 how to differentiate the THP-1 into macrophages is missing 

Line 219 how RIN is measured (by Bioanalyzer?), which kit?

2. some improper terms/phase, please check throughout the manuscripts

e.g. line 252 cell culture soup and line 279 give … cytokines

3. methods and results are mixed in the results section

PLOS authors have the option to publish the peer review history of their article (what does this mean?). If published, this will include your full peer review and any attached files.

Reviewer #1: No

Reviewer #2: No
---

## [Decision Letter · Decision Letter 1]

16 Nov 2023

Dear Dr. Prasad,

Thank you very much for submitting your manuscript "Taenia solium  excretory secretory proteins (ESPs) suppresses TLR4/AKT mediated ROS formation in human macrophages via hsa-miR-125" for consideration at PLOS Neglected Tropical Diseases. As with all papers reviewed by the journal, your manuscript was reviewed by members of the editorial board and by several independent reviewers. In light of the reviews (below this email), we would like to invite the resubmission of a significantly-revised version that takes into account the reviewers' comments. 

Your response to the reviewer comments on your original submission has not managed to clarify some major reviewer concerns, particularly regarding clarity of some of your experimental approaches. 

We cannot make any decision about publication until we have seen a revised manuscript and improved response to the original reviewers' comments. Your revised manuscript is also likely to be sent to reviewers for further evaluation.

Sincerely,

Andrew Scott MacDonald

Academic Editor

Cinzia Cantacessi

Section Editor

Reviewer's Responses to Questions

**Key Review Criteria Required for Acceptance?**

**Methods**

-Are the objectives of the study clearly articulated with a clear testable hypothesis stated?

-Is the study design appropriate to address the stated objectives?

-Is the population clearly described and appropriate for the hypothesis being tested?

-Is the sample size sufficient to ensure adequate power to address the hypothesis being tested?

-Were correct statistical analysis used to support conclusions?

-Are there concerns about ethical or regulatory requirements being met?

Reviewer #1: As previously stated, it is difficult to follow the experimental design of the authors. Some parts of the methods are not described either in the method section or in the figure legend (For example use of fMLP, or SC79 mentioned in the main result section but neither in figure nor in the material and methods). The text do not follow the order of the figures and those are not always cited in the text (For example which figure represent the SC79 treatment?) . It is incredibly complicated to review the article in those conditions and the authors have completely ignored my previous comment. 

Instead of clarifying the experimental design and the associated text of figure 5 and 6 as requested, the authors answered with a list of publications and a less than pleasant comment. 

Interestingly the paper cited, do indeed address my question and have the clear experimental design I was suggesting to the authors of the current manuscript to embrace. For example, Figure 1 of https://www.nature.com/articles/gene201438 actually compare LPS only versus LPS+worm extract versus worm extract only. Which is exactly what I was asking the authors with my previous comment.

Overall, what I can judge from the paper is of excellent quality and certainly worse reporting to the community, however ignoring comments from reviewer is extremely rude and certainly not acknowledging the time freely given to this review process.

Reviewer #2: (No Response)

**Results**

-Does the analysis presented match the analysis plan?

-Are the results clearly and completely presented?

-Are the figures (Tables, Images) of sufficient quality for clarity?

Reviewer #1: the clarity of the figures and associated results is not optimal

Reviewer #2: (No Response)

**Conclusions**

-Are the conclusions supported by the data presented?

-Are the limitations of analysis clearly described?

-Do the authors discuss how these data can be helpful to advance our understanding of the topic under study?

-Is public health relevance addressed?

Reviewer #1: Yes as much as I can judge

Reviewer #2: (No Response)

**Editorial and Data Presentation Modifications?**

Reviewer #1: Few typos : 

line 374 egligible 

line 416 capabilityp flouresence 

Figures are often distorted, probably during reduction

Reviewer #2: (No Response)

**Summary and General Comments**

Reviewer #1: (No Response)

Reviewer #2: (No Response)

PLOS authors have the option to publish the peer review history of their article (what does this mean?). If published, this will include your full peer review and any attached files.

Reviewer #1: No

Reviewer #2: No
---

## [Editor Report · Decision Letter 2]

12 Dec 2023

Dear Dr. Prasad,

We are pleased to inform you that your manuscript 'Taenia solium  excretory secretory proteins (ESPs) suppresses TLR4/AKT mediated ROS formation in human macrophages via hsa-miR-125' has been provisionally accepted for publication in PLOS Neglected Tropical Diseases.

Best regards,

Andrew Scott MacDonald

Academic Editor

Cinzia Cantacessi

Section Editor

---

## [Editor Report · Acceptance letter]

20 Dec 2023

Dear Dr. Prasad,

We are delighted to inform you that your manuscript, " Taenia solium  excretory secretory proteins (ESPs) suppresses TLR4/AKT mediated ROS formation in human macrophages via hsa-miR-125 ," has been formally accepted for publication in PLOS Neglected Tropical Diseases.

Best regards,

Shaden Kamhawi

co-Editor-in-Chief

Paul Brindley

co-Editor-in-Chief
